# *KDM6A* Variants Increased Relapse Risk in Adult Acute Myeloid Leukemia

**DOI:** 10.3390/cancers17132236

**Published:** 2025-07-03

**Authors:** Yijing Zhao, Liting Niu, Sen Yang, Lu Yu, Ting Zhao, Hao Jiang, Lanping Xu, Yu Wang, Xiaohui Zhang, Xiaojun Huang, Qian Jiang, Feifei Tang

**Affiliations:** 1Peking University People’s Hospital, Peking University Institute of Hematology, National Clinical Research Center for Hematologic Disease, Beijing Key Laboratory of Cell and Gene Therapy for Hematologic Malignancies, Peking University, Beijing 100044, China; 2Peking-Tsinghua Center for Life Sciences, Academy for Advanced Interdisciplinary Studies, Peking University, Beijing 100044, China; 3Department of Hematology, Peking University People’s Hospital, Qingdao 266111, China

**Keywords:** acute myeloid leukemia, *KDM6A*, *RUNX1::RUNX1T1*

## Abstract

*KDM6A* (lysine demethylase 6A) is an epigenetic modulator involved in chromatin remodeling and gene expression. However, the impact of *KDM6A* variants on the cumulative incidence of relapse in adults with AML in histological remission is untested. This study validates *KDM6A* mutations as rare but recurrent in AML, particularly in *RUNX1::RUNX1T1* AML, where they predict poor outcomes and high relapse risk. Early molecular relapse and resistance to conventional therapies highlight the need for frequent monitoring and targeted interventions. These findings underscore the potential of epigenetic therapies and emphasize the importance of *KDM6A* as a prognostic biomarker, paving the way for improved management of AML patients.

## 1. Introduction

Acute myeloid leukemia (AML) is a heterogeneous group of aggressive hematologic malignancies driven by different genetic and epigenetic aberrations. These mutations may promote leukemic cell proliferation, impair differentiation, and enhance cell survival, making AML a particularly challenging disease to treat [1]. Although extensive research has identified numerous genetic drivers, ongoing studies continue to uncover the role of less common mutations and their interactions, which may offer therapeutic potential.

*KDM6A* (lysine demethylase 6A) is an epigenetic modulator involved in chromatin remodeling and gene expression [2]. Variants of *KDM6A* result in global epigenetic dysregulation and are associated with diverse cancers, including acute myeloid leukemia (AML). Pre-clinical data indicate that *KDM6A* variants are associated with a worse survival in AML, possibly because of repressive H3K27me3 marks, which silence genes critical for hematopoietic differentiation [3,4,5]. However, the impact of *KDM6A* variants on the cumulative incidence of relapse in adults with AML in histological remission is untested.

This study aimed to investigate the correlation between *KDM6A* mutations and relapse risk in AML. Our preliminary research found that *KDM6A* mutations predicted poor outcomes in patients with *RUNX1::RUNX1T1* [6], so we conducted a subgroup analysis on the *RUNX1::RUNX1T1* subtype. By integrating genomic data, measurable residual disease (MRD) information, and clinical outcomes, we assessed the impact of *KDM6A* variants on the cumulative incidence of relapse in 1676 adults with AML in histological remission, including a subgroup of 207 patients with *RUNX1::RUNX1T1* fusion.

## 2. Materials and Methods

### 2.1. Participants

Data from 1970 consecutive adult patients with AML diagnosed and treated between January 2017 and July 2024 at Peking University People’s Hospital were reviewed. AML was diagnosed by histology, immunology, cytogenetics, and genetic abnormalities as described. The study enrolled participants who met the following inclusion criteria: (1) age ≥ 16 years; (2) histological complete remission (CR) achieved after induction therapy. Among 1970 AML patients, 1676 (85.1%) achieved final CR/CRi after induction chemotherapy and were enrolled in this study. A total of 207 participants with the *RUNX1::RUNX1T1* fusion gene were eligible, along with 178 participants with the *RUNX1::RUNX1T1* quantitation and MRD reduction record. This study was approved by Peking University People’s Hospital Ethics Committee (2025PHB305-001); conducted as per the Declaration of Helsinki.

### 2.2. Diagnosis, Monitoring, and Therapy Responses

Diagnosis, monitoring, and treatment responses adhered to the 2022 European Leukemia Net (ELN) recommendations [7]. Immune phenotyping was performed using multi-parameter flow cytometry with CD45/side scatter (SSC) gating. Cytogenetic analyses were conducted using standard G-banding techniques. Molecular screening for leukemia-associated fusion genes and high-depth targeted regional sequencing (TRS) was performed for all patients. Demographic and clinical variables, including complete blood count (CBC) and results of initial hematological, cytogenetic, and molecular analyses, were extracted from medical records. Measurable residual disease was identified using real-time quantitative polymerase chain reaction (RT-qPCR) and multi-parameter flow cytometry measured minimal residual disease (MRD). RT-qPCR was performed on BM samples using DNA extracted with DNAzol kits (Invitrogen, Carlsbad, CA, USA) following standard protocols as previously described [8]. Primers and TaqMan^®^ probes (Foster City, CA, USA) for the target gene and internal control (ALB) were designed with Primer Express 2.0. Reactions were run on an ABI PRISM^®^ 7500 (Foster City, CA, USA) using TaqMan^®^ Universal PCR Master Mix (Foster City, CA, USA), with specific primer/probe concentrations and 150–250 ng DNA. Cycling conditions included an initial step at 50 °C for 2 min and 95 °C for 10 min, followed by 50 cycles of 95 °C for 15 s and 60 °C for 1 min. Gene expression was calculated by normalizing target gene copies to ALB. Detection sensitivity ranged from 10^−4^ to 10^−5^ [8]. In *RUNX1::RUNX1T1* AML, molecular response (MR)^2.5^ (>2.5 log reduction) was determined after treatment cycle 1, while MR^3.0^ (>3.0 log reduction) was identified after treatment cycle 2, according to the MRD detection guideline in previous study [9].

The induction regimens, as detailed in previous studies [6], comprised intensive therapy options including the homoharringtonine-cytarabine-aclarubicin (HAA) regimen and the idarubicin-cytarabine (IA) regimen, and the DA regimen. For patients unsuitable for intensive therapy, less intensive approaches such as hypomethylating agents (HMAs) combined with venetoclax (VEN) or the CAG regimen were employed. For favorable risk AML, patients who achieved CR or CR with incomplete hematologic recovery (CRi) received high-dose cytarabine-based consolidation therapy for 3–4 cycles or less intensive continued therapy. If patients did not achieve CR/CRi after 2 cycles of induction or relapsed, intermediate or high-dose cytarabine-based regimens, such as revised CLAG (cladribine, cytarabine, and granulocyte colony-stimulating factor [G-CSF]) or FLAG (fludarabine, cytarabine and G-CSF) regimens, were considered as salvage therapy. For intermediate and adverse risk AML, patients eligible for allogeneic hematopoietic stem cell transplantation (allo-HSCT) underwent ≥ 2 cycles of consolidation chemotherapy. If transplantation is not eligible, the chemotherapy regimen will be determined by the physician [10]. Donor selection included human leukocyte antigen (HLA)-matched siblings, HLA-matched unrelated donors, or HLA haploidentical-related donors. Allo-HSCT was performed following previously reported methodologies. After the second consolidation course, eligible participants with a potential donor and physicians discussed the risks and benefits of transplant versus continuing consolidation chemotherapy, considering covariates, such as risk stratification, measurable residual disease test results, economics, and patient preference. Based on these discussions, 735 of 1676 participants (43.8%) and 78 of 207 *RUNX1::RUNX1T1* fusion-positive participants (37%) received a transplant after a median of 4 (2–6) courses of consolidation chemotherapy, as described.

### 2.3. High-Depth Targeted Regional Sequencing (TRS)

TRS was performed on bone marrow samples from patients initially diagnosed with AML, and sequencing was conducted by Kingmed Diagnostics in Guangzhou. The deep-targeted sequencing panel initially comprised 175 genes for patients diagnosed between 2018 and 2020, and it was expanded to 290 genes for patients diagnosed from 2021 onwards. All genes included in the targeted sequencing panel were associated with hematological myeloid malignancies (Table A1 in Appendix A). DNA sequencing was executed using the Illumina NovaSeq6000 system (Illumina, San Diego, CA, USA) in accordance with the manufacturer’s recommendations. Variant curation adhered to the Standards and Guidelines for the Interpretation and Reporting of Sequence Variants in Cancer. This study primarily focused on variants categorized as having strong clinical significance (Tier I) and those presenting potential clinical significance (Tier II) [11].

### 2.4. Statistical Analyses

Propensity score matching (PSM) was employed to estimate the causal effect of *KDM6A* mutations on clinical outcomes. Among the 1677 participants, covariates, such as age, gender, ELN 2022 risk category [7], Eastern Cooperative Oncology Group (ECOG) performance status, and whether allo-HSCT was performed after CR1 were included in the analysis. The matching procedure followed a 1:10 strategy, where 27 participants with *KDM6A* mutations were matched with 270 participants with the wild-type genotype. The most frequent chromosomal abnormality co-occurring with *KDM6A* mutations was *RUNX1::RUNX1T1* fusion, observed in 48.1% of cases (*n* = 13). For the *RUNX1::RUNX1T1* fusion AML, additional covariates, including age, sex, MR^2.5^, MR^3.0^, and *KIT* mutation, and allo-HSCT status, were incorporated [12]. A similar 1:5 matching strategy was applied, with each treated participant matched to 10 control participants using nearest-neighbor matching with a 0.2 caliper. The balance of covariates before and after matching was assessed using standardized mean differences (SMDs) to ensure comparability between groups.

Descriptive statistics were employed to summarize covariates, with categorical variables presented as counts and percentages and continuous variables expressed as medians and interquartile ranges (IQRs). The Pearson chi-square test was applied to analyze categorical covariates. The correlation analysis was performed using the Pearson correlation coefficient to assess the linear relationship between genes. In contrast, the Student’s *t*-test (for normal distributed data) or the Mann–Whitney U test (for non-normally distributed data) was used to assess continuous covariates. Cox regression models were used to conduct multivariable analyses to identify covariates associated with overall survival (OS) and relapse-free survival (RFS). Variance inflation factor (VIF) was estimated to assess multicollinearity among covariates in the Cox model. OS and RFS were calculated using the Kaplan–Meier method with the log-rank test. The CIR was assessed using competing risk analysis, and Gray’s test compared differences between groups. A two-sided *p*-value < 0.05 was considered significant. For analysis and graphing, SPSS 27.0 (SPSS, Chicago, IL), R version 4.0.2 (R Core Team, Vienna, Austria), and GraphPad Prism 10 (GraphPad Software Inc., Boston, MA, USA) were employed.

### 2.5. Bioinformatics Analysis

Bioinformatics analyses were performed using R. Mutation annotation and visualization were conducted with the maftools package, including lollipop plots for *KDM6A* variants. Gene co-mutation patterns were assessed using corrplot and visualized with circlize and ComplexHeatmap. Baseline clinical characteristics were summarized using the tableone package. Data processing and figure generation were supported by the tidyverse suite.

## 3. Results

### 3.1. Study Participant Demographics

A total of 1676 consecutive patients with AML were enrolled in this study. The study flowchart is shown in Figure 1. After performing a (1:10) PSM for *KDM6A* mutations, 297 participants were selected for analysis (mutation: wild-type, 27:270), with key demographical and clinical data summarized in Table 1. Among the 297 participants, 204 (68.7%) were female. The median age was 45 years (IQR 33–57 years). The risk stratification based on the 2022 ELN classification identified 126 patients (42.4%) in the favorable risk category, 80 (26.9%) in the intermediate risk category, and 91 (30.6%) in the adverse risk category. All 297 participants received induction therapy, with the following distribution: 86 (29%) received HAA; 95 (32%) received IA; 19 (6.4%) received DA; 48 (18.2%) were treated with venetoclax and azacitidine; 29 (9.8%) received CAG; and 14 (4.7%) received unclassifiable therapy. Of these participants, 231 (77.8%) achieved CR/CRi following the first induction chemotherapy. Among them, 88 (29.6%) underwent allo-HSCT during CR1. For the 297 participants, the median follow-up time for survival was 1.5 years (IQR, 0.7–2.7). During this period, 26 patients (14.6%) died. The median OS and RFS were 1.5 years (IQR, 0.7–2.7) and 1.3 years (IQR, 0.5–2.4), respectively. A total of 104 patients (35.0%) relapsed, with the median relapse time being 0.68 years (IQR, 0.3–1.4).

### 3.2. Mutation Analysis Overview

*KDM6A* mutations occurred in 27 (1.6%) patients across all AML subtypes. The OncoPrint visualization showed the mutation landscape, co-occurrence patterns, chromosomal abnormalities, and clinical information in *KDM6A*-mutated AML patients in Figure 2A. As shown by Figure 2A,B, the most frequent chromosomal abnormality co-occurring with *KDM6A* mutations was the *RUNX1::RUNX1T1* fusion genes mutations, found in 48.1% of cases (*n* = 13). The most co-occurring mutations were the *KIT* gene (8 patients, 29.6%) and *ASXL1* (7 patients, 25.9%), followed by *RUNX1* (5 patients, 18.5%), *TET2* (5 patients, 18.5%), and *NRAS* (5 patients, 18.5%), which may have synergistic effects on disease progression. Based on our further investigation into the co-occurrence relationship between *KDM6A* and other genes, of 27 *KDM6A* mutations, 13 (48.1%) co-occurred with the *RUNX1::RUNX1T1* fusion (r = 0.25, *p* < 0.001), revealing a significant association between *KDM6A* and *RUNX1::RUNX1T1* fusion genes (Figure 2C). *KDM6A* mutations did not show significant co-occurrence with *FLT3*-ITD, *FLT3*-TKD, *CEBPA*, or any other mutations in our cohort. 

### 3.3. Clinical Impact of KDM6A Mutations

With respect to clinical outcomes, the *KDM6A* mutation was associated with shorter relapse-free survival, with a 2-year RFS of 60.1% for the wild-type group versus 34.7% for the mutation group (*p* = 0.016) in univariable analyses of 1676 consecutive patients with AML achieving CR/CRi (Figure 3A). The *KDM6A* mutation had a significantly higher 2-year CIR compared with subjects with wild-type (50.9% [54.6%, 47.3%] versus 33.8% [33.7%, 33.8%], *p* = 0.06 (Figure 3G). In the Fine–Gray regression, the *KDM6A* mutation was independently associated with an increased relapse risk (HR = 1.72 [1.01–2.94], *p* = 0.05). After PSM in 297 participants, the *KDM6A* mutation was also associated with a shorter RFS, with a 2-year RFS of 60.9% v.s. 36.3% (p = 0.044) (Figure 3B). The *KDM6A* mutation had a significantly higher 2-year CIR compared with subjects with wild-type (45.7% [41.6%, 49.7%] versus 28.6% [28.2%, 29.0%], *p* = 0.04) (Figure 3H). In the Fine–Gray regression, the *KDM6A* mutation was independently associated with an increased relapse risk (HR = 1.98 [1.08–3.63], *p* = 0.03). Likewise, the *KDM6A* mutation was an independent prognostic factor in multivariable analysis for RFS (3.078 [1.56–6.08]; *p* = 0.001) (Table 2). No significant difference was observed in OS, with 2-year OS rates of 65.9% v.s. 72.1% (*p* = 0.249). Multivariable analyses confirmed that this variable was not an independent prognostic factor for OS (HR = 1.82, [0.84–3.95]; *p* = 0.131).

### 3.4. KDM6A Mutations in RUNX1::RUNX1T1 AML

Since approximately half of the *KDM6A* mutations co-occurred with the *RUNX1::RUNX1T1* fusion gene, we further analyzed the *RUNX1::RUNX1T1* AML in patients harboring the *KDM6A* mutation. A total of 207 participants with the *RUNX1::RUNX1T1* fusion gene were deemed eligible for analysis, of whom 106 (51%) were men. The median age was 37 years (IQR: 27–48.5 years). According to the 2022 ELN risk classification, all participants were classified as low risk. Following PSM (1:5.8) for *KDM6A* mutation, 81 participants from the *RUNX1::RUNX1T1* fusion gene–AML cohort were selected for analysis. Baseline characteristics are summarized in Table 3.

We further detected the mutation landscape of *RUNX1::RUNX1T1* AML in patients harboring the *KDM6A* mutation, with a median of six mutations per participant (0–13). In *RUNX1::RUNX1T1* AML patients, *KIT* was the most frequently mutated gene (81 patients, 46%), followed by *ASXL2* (32 patients, 17%) and *ASXL1* (28 patients, 16%). *KDM6A* mutations were detected in 13 (6%) patients with the *RUNX1::RUNX1T1* fusion gene–AML, compared with 27 (1.6%) across all AML subtypes. Most identified mutations were frameshift insertions, with a total of nine such instances observed (Figure A1A). The median variant allele frequency (VAF) for these mutations was 36% (2.3–91.1%). Notably, 10 mutations exhibited a VAF exceeding the 10% threshold.

Referring to clinical outcomes, similarly, in the *RUNX1::RUNX1T1* fusion gene subgroup, the *KDM6A* mutation was also associated with a shorter RFS, with a 2-year RFS of 27.7% vs. 75.8% (*p* < 0.001) and OS, and with a 2-year OS of 66.7% vs. 86.2% (*p* = 0.003) in univariable analyses (Figure 3C,D). Likewise, the *KDM6A* mutation was an independent prognostic factor in multivariable analyses for RFS (5.1 [2.5–10.5]; *p* < 0.001) and OS (12.6 [4.3–38.7]; *p* < 0.001) (Table 4).

In the *RUNX1::RUNX1T1* fusion gene PSM subgroup, the *KDM6A* mutation was also associated with a shorter RFS, with a 2-year cumulative RFS of 31.3% vs. 71.9% (*p* < 0.001), and OS with a 2-year cumulative OS of 63.6% vs. 86.4% (*p* = 0.002) in univariable analyses (Figure 3E,F). The *KDM6A* mutation had a significantly higher 2-year CIR compared with subjects with wild-type (46.9% [41.7, 52.1%] versus 24.0% [22.8, 25.1%], *p* = 0.05) (Figure 3I). In the Fine–Gray regression, the *KDM6A* mutation was independently associated with an increased relapse risk (HR = 2.46 [1.11–5.47], *p* = 0.03). Likewise, the *KDM6A* mutation was an independent prognostic factor in multivariable analyses for RFS (5.7 [1.1–10.3]; *p* = 0.039), and OS (8.7 [1.2–16.2]; *p* = 0.026) (Table 5). These findings suggest that *KDM6A* mutations are associated with poor prognosis in AML patients, especially the *RUNX1::RUNX1T1* fusion gene subgroup.

We conducted a retrospective analysis of MRD monitoring in 13 AML patients with *KDM6A* mutations and the *RUNX1::RUNX1T1* fusion gene. We observed that these patients experienced varying degrees of molecular relapse after achieving remission. The average time to molecular relapse, determined using RT-PCR detection of *RUNX1::RUNX1T1* transcripts, was 43 d (IQR: 53–60) after the deepest molecular remission. Figure A1C depicts the molecular monitoring results, showing the time course of molecular relapse for each patient. Most patients exhibited a detectable rise in *RUNX1::RUNX1T1* fusion gene transcripts within the first two months after achieving remission.

## 4. Discussion

Based on the large scale analyzed for this alteration, we validated *KDM6A* mutations as a rare but recurrent genetic lesion in AML (1.6%), and *RUNX1::RUNX1T1* AML (6.3%). *KDM6A* mutations are an independent prognostic marker for poor clinical outcomes, associated with a specific co-occurrence profile with the *RUNX1::RUNX1T1* fusion gene.

To our knowledge, this study represents one of the first comprehensive investigations examining the clinical impact of *KDM6A* mutations in patients with AML. Despite emerging evidence implicating *KDM6A* alterations in hematologic malignancies [13], prior research has predominantly focused on its transcriptional downregulation or epigenetic dysregulation rather than somatic mutations. Notably, only one previous study from our institute reported an association between *KDM6A* mutations and adverse prognosis in the subset of *RUNX1::RUNX1T1*-positive CBF-AML, partially aligning with our findings [6]. However, that study did not explore the broader implications of *KDM6A* mutations across unselected AML cohorts, leaving a critical gap in understanding their all subtype of AML relevance. 

While several groups have linked reduced *KDM6A* expression to chemotherapy resistance and poor outcomes [2,14,15,16], these observations operate at a distinct mechanistic level compared to genomic *KDM6A* mutations. *KDM6A* promotes AML progression and drug resistance through its tumor suppressor functions and involvement in DNA repair mechanisms [17]. *KDM6A* is frequently mutated in AML, leading to a loss of function and increased drug resistance [5]. For example, *KDM6A* mutations are associated with higher IC50 values for cytarabine, indicating reduced sensitivity to this common AML treatment. Additionally, *KDM6A* regulates the expression of DNA repair genes. Its deficiency impairs the DNA damage response (DDR), leading to increased sensitivity to PARP and BCL2 inhibition [4]. In AML cells, *KDM6A* is recruited to the transcriptional start sites of key homologous recombination genes upon DNA damage, facilitating their transcription. Loss of *KDM6A* activity, either through genetic or pharmacological inhibition, results in elevated H3K27me3 levels at these regulatory elements, preventing gene transcription and compromising DNA repair. This highlights *KDM6A*’s essential role in maintaining genomic stability and its potential as a therapeutic target in AML. Expression level-based studies reflect regulatory or epigenetic perturbations, whereas truncating or loss-of-function mutations directly impair *KDM6A*’s function, potentially exacerbating genomic instability and altering transcriptional programs in a mutation-specific manner [3,4,5]. Previous research has primarily focused on the impact of the *KDM6A* expression level on tumor recurrence and treatment resistance, with limited direct data on complete remission rates [18]. *KDM6A* mutations may enhance tumor cell drug resistance or promote immune evasion, thereby affecting treatment outcomes and recurrence rates [19].

Our data extend these prior observations by demonstrating that *KDM6A* mutations, independent of expression changes, confer a high-risk phenotype characterized by shortened survival, even after adjusting for established prognostic factors such as age, cytogenetics, MRD, and ELN guidelines for AML risk stratification. The uniqueness of our study lies in its systematic integration of clinical, molecular, and functional data to establish *KDM6A* mutations as independent biomarkers of adverse prognosis. In AML, there are few studies on the coexistence of *KDM6A* mutations with other gene mutations or chromosomal abnormalities. To date, no large-scale AML cohorts have specifically interrogated the genetic signature, prognostic, or therapeutic relevance of *KDM6A* mutations, underscoring the novelty of our findings.

Our study had some limitations. First, it was a retrospective single-center clinical study, which inherently involves therapy selection biases. Second, we lacked functional studies to validate the clinical findings. Despite these limitations, the findings represent an important step towards understanding the *KDM6A* mutations influencing the prognosis of AML and could inform future therapeutic strategies.

## 5. Conclusions

In conclusion, our study validates *KDM6A* mutations as rare but recurrent in AML, particularly in *RUNX1::RUNX1T1* AML, where they predict poor outcomes and a high relapse risk. Early molecular relapse and resistance to conventional therapies highlight the need for frequent monitoring and targeted interventions. These findings underscore the potential of epigenetic therapies and emphasize the importance of *KDM6A* as a prognostic biomarker, paving the way for improved management of AML patients.

## Figures and Tables

**Figure 1 cancers-17-02236-f001:**
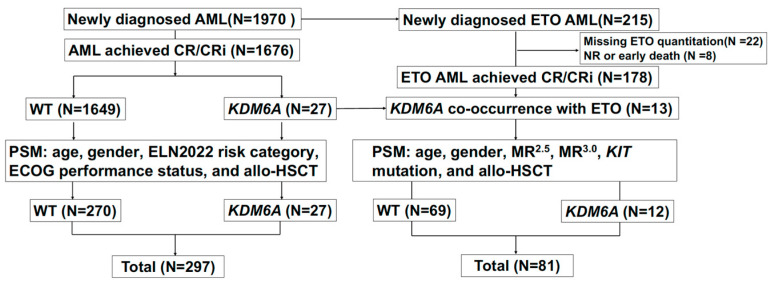
Flowchart.

**Figure 2 cancers-17-02236-f002:**
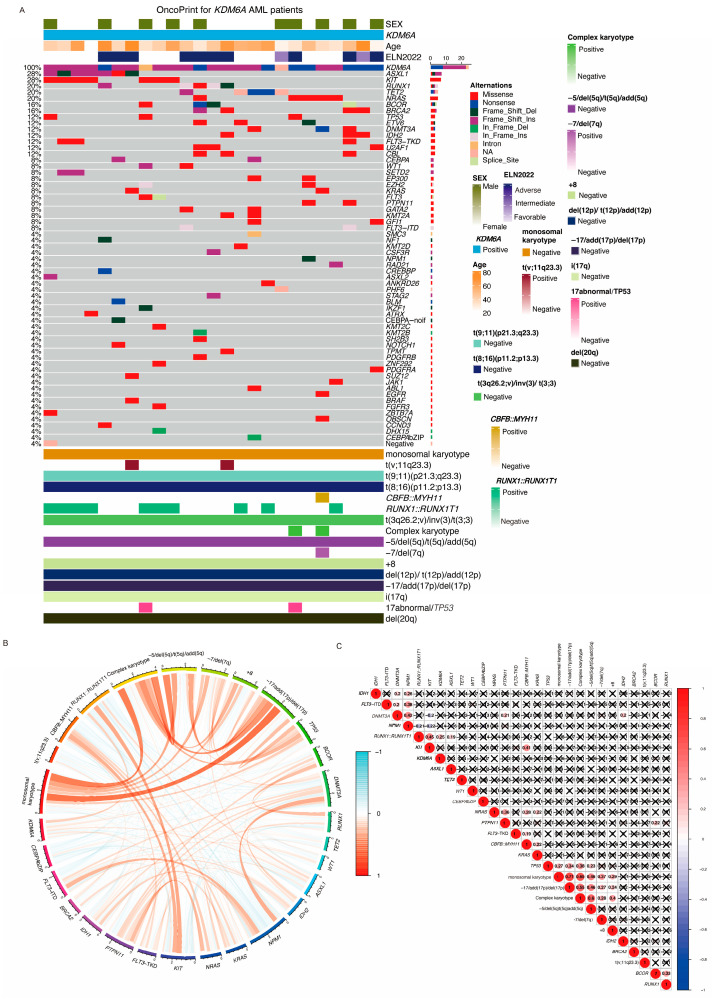
*KDM6A* OncoPrint visualization. (**A**) OncoPrint of *KDM6A*-mutated AML patients. Each row represents an individual patient, and colored blocks indicate different genetic alterations. The top panel shows patient characteristics. The leftmost column lists mutated genes, with mutation types annotated by different colors. The right panel summarizes key cytogenetic abnormalities and their presence (positive) or absence (negative) in these patients. *CEBPA*-noif: *CEBPA* bZIP zone mutation but not in frame mutation; *CEBPA* bZIP: bZIP zone in frame mutation; *CEBPA*: *CEBPA* mutation out of bZIP zone. Mutation (**B**) Circos plot of co-occurring genetic alterations. This circular diagram visualizes the co-occurrence and mutual exclusivity of genetic alterations in *KDM6A*-mutated AML patients. Orange lines represent co-occurring alterations, while blue lines indicate mutually exclusive events. Different genetic events, including chromosomal abnormalities and gene mutations, are labeled on the outer circle. (**C**) Correlation matrix of genetic alterations. A heatmap illustrating the correlation between different genetic alterations. Red circles indicate positive correlations, while blue circles denote negative correlations. The size and intensity of the circles correspond to the strength of the correlation. Black “X” marks indicate non-significant correlations.

**Figure 3 cancers-17-02236-f003:**
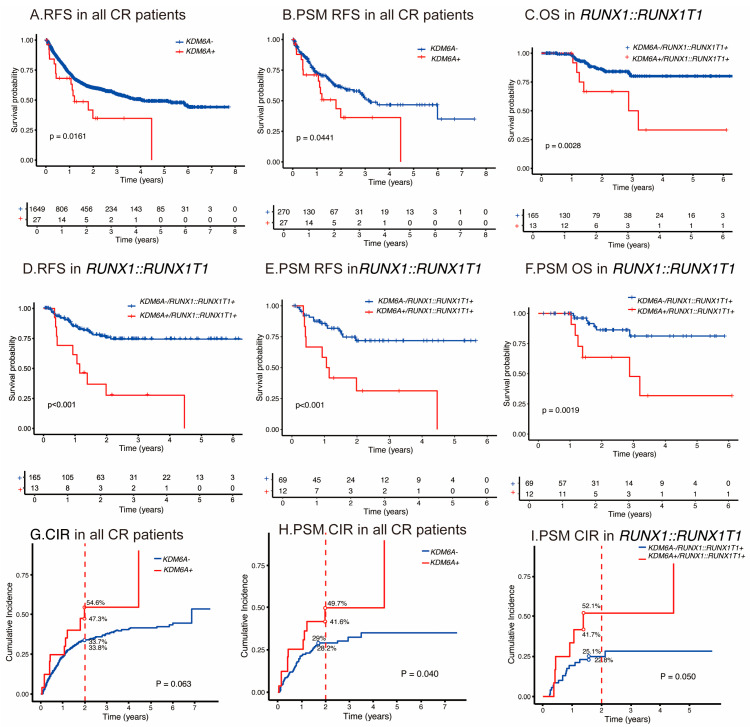
Outcomes of *KDM6A* mutations for AML. (**A**) RFS in cohorts with wild-type or *KDM6A* mutated group in all AML subgroups. (**B**) RFS after PSM in cohorts with wild-type or *KDM6A* mutations across all AML risk categories. (**C**) OS in cohorts with wild-type or *KDM6A* mutations in *RUNX1::RUNX1T1* fusion gene–AML. (**D**) RFS in cohorts with wild-type or *KDM6A* mutated group in *RUNX1::RUNX1T1* fusion gene–AML. (**E**) RFS after PSM in cohorts with wild-type or *KDM6A* mutated group in *RUNX1::RUNX1T1* fusion gene–AML. (**F**) OS after PSM in cohorts with wild-type or *KDM6A* mutated group in *RUNX1::RUNX1T1* fusion gene–AML. CIR for *KDM6A* mutations. (**G**–**I**) according to the all-risk category AML, PSM in all-risk category, and in *RUNX1::RUNX1T1* fusion gene–AML.

**Table 1 cancers-17-02236-t001:** Baseline characteristics of the 297-propensity score-matched patients who achieved CR/CRi.

	Level	Overall	Wild-type	*KDM6A* Mutation	*p*
n		297	270	27	
Sex (%)	Female	204 (68.69)	187 (69.26)	17 (62.96)	0.65
	Male	93 (31.31)	83 (30.74)	10 (37.04)	
Age (median [IQR])		45.00 [33.00, 57.00]	46.50 [33.00, 56.75]	44.00 [33.50, 59.00]	0.86
ECOG (%)	0	187 (62.96)	168 (62.22)	19 (70.37)	0.61
	1	87 (29.29)	80 (29.63)	7 (25.93)	
	2	23 (7.74)	22 (8.15)	1 (3.70)	
ELN2022 (%)	Favorable	126 (42.42)	112 (41.48)	14 (51.85)	0.05
	Intermediate	80 (26.94)	78 (28.89)	2 (7.41)	
	Adverse	91 (30.64)	80 (29.63)	11 (40.74)	
Induction chemotherapy (%)	HAA	86 (28.96)	76 (28.15)	10 (37.04)	0.61
	IA	95 (31.99)	89 (32.96)	6 (22.22)	
	DA	19 (6.40)	16 (5.93)	3 (11.11)	
	VEN + AZA	48 (18.18)	45 (18.89)	3 (11.11)	
	CAG	29 (9.76)	25 (9.26)	4 (14.81)	
	Unclassifiable	14 (4.71)	13 (4.81)	1 (3.70)	
Allo-HSCT at CR1 (%)	No	209 (70.37)	190 (70.37)	19 (70.37)	1
	Yes	88 (29.63)	80 (29.63)	8 (29.63)	
CR/CRi at 1st induction chemotherapy (%)	No	66 (22.22)	61 (22.59)	5 (18.52)	0.81
	Yes	231 (77.78)	209 (77.41)	22 (81.48)	
Relapse (%)	No	193 (64.98)	180 (66.67)	13 (48.15)	0.09
	Yes	104 (35.02)	90 (33.33)	14 (51.85)	
Dead (%)	No	225 (75.76)	207 (76.67)	18 (66.67)	0.36
	Yes	72 (24.24)	63 (23.33)	9 (33.33)	

**Table 2 cancers-17-02236-t002:** Multivariate Cox analyses determining the prognostic significance of *KDM6A* after PSM in all AML subgroups.

RFS Variable	*p*	HR (95% CI)
*KDM6A*	0.001	3.078 (1.56–6.08)
*BCOR*	0.052	0.23 (0.05–1.01)
MRD (after 1st cycle treatment FCM)	0.088	1.52 (0.94–2.47)
*CSF1R*	0.007	4.13 (1.47–11.58)
*INPP5D*	0.03	9.94 (1.25–78.95)
ELN2022	0.041	
Favorable	reference	reference
Intermediate	0.261	1.41 (0.77–2.59)
Adverse	0.012	2.10 (1.18–3.73)

**Table 3 cancers-17-02236-t003:** Baseline characteristics of the 81 propensity-score-matched patients who achieved CR/CRi with the *RUNX1::RUNX1T1* fusion gene–AML.

	Overall	Wild-Type	*KDM6A* Mutation	*p*
n	81	69	12	
Sex (%) Male	41 (50.6)	36 (52.2)	5 (41.7)	0.7
Age	38 [27–52]	38 [27–51]	40 [30.3–55.8]	0.9
PB blast	40 [27–50]	39 [27–50]	48 [35.8–60.8]	0.1
BM blast	46 [32–60]	44 [32–60]	51.5 [45.5–68.6]	0.2
WBC (×10^9^/L)	8.1 [5.1–16.2]	8.1 [4.9–16.1]	10.7 [6.3–28.8]	0.3
HB (g/L)	79 [62–96]	79 [60–100]	81.8 [63–89.8]	0.8
PLT (×10^9^/L)	29 [21–43]	29 [16–42]	30 [23.5–66.1]	0.3
ASXL1 (%)	16 (19.8)	12 (17.4)	4 (33.3)	0.4
KIT (%)	40 (49.4)	34 (49.3)	6 (50)	1
CSF3R (%)	6 (7.4)	6 (8.7)	0 (0)	0.6
NRAS (%)	12 (14.8)	11 (15.9)	1 (8.3)	0.8
SETD2 (%)	3 (3.7)	1 (1.5)	2 (16.7)	0.1
KRAS (%)	4 (4.9)	4 (5.8)	0 (0)	0.9
ASXL2 (%)	15 (18.5)	14 (20.3)	1 (8.3)	0.6
Mutation number	2 [1–3]	2 [1–3]	2.5 [1.8–5]	0.2
Dead (%)	13 (16.1)	7 (10.1)	6 (50)	<0.01
Relapse (%)	24 (29.6)	15 (21.7)	9 (75)	<0.01
Allo-HSCT (%)	33 (40.7)	29 (42)	4 (33.3)	0.8
Achieved 1st cycle MR2.5 (%)	50 (61.7)	44 (63.8)	6 (50)	0.6
Achieved 2nd cycle MR3.0 (%)	33 (40.7)	30 (43.5)	3 (25)	0.4
Achieved consolidation 2nd cycle MR3.0 (%)	1	43 (54.4)	37 (54.4)	1

**Table 4 cancers-17-02236-t004:** Multivariate analyses determining the prognostic significance of *KDM6A*.

**RFS**	** *p* **	**HR (95% CI)**
*KDM6A*	<0.001	5.1 (2.5–10.5)
Fail to achieve MR^3.0^	0.004	2.5 (1.3–4.7)
**OS**	** *p* **	**HR (95% CI)**
*KDM6A*	<0.001	12.6 (4.3–38.7)
Fail to achieve MR^3.0^	0.002	6.4 (2.0–20.6)

**Table 5 cancers-17-02236-t005:** Multivariate Cox analyses determining the prognostic significance of *KDM6A* after PSM.

**RFS Variable**	** *p* **	**HR (95% CI)**
*KDM6A*	0.039	5.7 (1.1–10.3)
AGE	0.254	1 (1–1.1)
*ASXL1*	0.294	1.1 (0.1–2)
Fail to achieve MR^3.0^	0.453	1.8 (0.6–3.1)
SEX (male)	0.688	1.2 (0.3–2.1)
*KIT*	0.733	1.5 (0.5–2.6)
Fail to achieve MR^2.5^	0.892	1.9 (0.3–3.4)
**OS Variable**	** *p* **	**HR (95% CI)**
AGE	0.003	1 (1.0–1.1)
*ASXL1*	0.014	0.3 (0.01–0.6)
*KDM6A*	0.026	8.7 (1.2–16.2)
*KIT*	0.034	4.3 (1.08–7.5)
Fail to achieve MR^3.0^	0.114	3.3 (0.83–5.8)
Fail to achieve MR^2.5^	0.526	4.7 (0.32–9.1)
SEX (male)	0.979	1.5 (0.37–2.7)

## Data Availability

Data are contained within the article.

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
