# Peer review of "KDM6A Variants Increased Relapse Risk in Adult Acute Myeloid Leukemia"

_cancers, 2025, doi:10.3390/cancers17132236_

Round 1
Reviewer 1 Report
Comments and Suggestions for Authors
The manuscript focused on evaluating the impact of KDM6A mutations on relapse risk in adult AML patients. The study was well designed. The results are impressive with a small KDM6A mutated cohort (27 out of 1676 patients), showing a significant correlation between KDM6A mutations and relapse risk and poor prognosis. Interestingly KDM6A mutations are rare, only occur in 2-5% of AML but demonstrate such a strong association with relapse risk and poor prognosis, especially in patients with RUNX1::RUNX1T1 fusion, with what authors have demonstrated in the manuscript.
Introduction:
Need more background information: why the study was specifically focused on the RUNX1::RUNX1T1 subtype? RUNX1::RUNX1T1 subtype occurs in 5-10% of adult AML patients. What is the rationale of focusing on this subtype? Authors addressed that “This study aimed to investigate the correlation between KDM6A mutations and relapse risk in AML, with a specific focus on RUNX1::RUNX1T1 subtype” (line 58-59), but the reason for this specific focus is unclear and there is no background introduction of how KDM6A mutations link with RUNX1::RUNX1T1 subtype. Please provide sufficient background info to highlight the significance of the study.
Discussion:
Line 283-285: Please remove. The guidelines of how and what to write for discussion don’t need to be shown in Discussion.
Author Response
Reviewer 1 Comments
The manuscript focused on evaluating the impact of KDM6A mutations on relapse risk in adult AML patients. The study was well designed. The results are impressive with a small KDM6A mutated cohort (27 out of 1676 patients), showing a significant correlation between KDM6A mutations and relapse risk and poor prognosis. Interestingly KDM6A mutations are rare, only occur in 2-5% of AML but demonstrate such a strong association with relapse risk and poor prognosis, especially in patients with RUNX1::RUNX1T1 fusion, with what authors have demonstrated in the manuscript.
Comments 1: Introduction:Need more background information: why the study was specifically focused on the RUNX1::RUNX1T1 subtype? RUNX1::RUNX1T1 subtype occurs in 5-10% of adult AML patients. What is the rationale of focusing on this subtype? Authors addressed that “This study aimed to investigate the correlation between KDM6A mutations and relapse risk in AML, with a specific focus on RUNX1::RUNX1T1 subtype” (line 58-59), but the reason for this specific focus is unclear and there is no background introduction of how KDM6A mutations link with RUNX1::RUNX1T1 subtype. Please provide sufficient background info to highlight the significance of the study.
Response 1: Thank you for pointing this out. We have now clarified the rationale for our focus on the RUNX1::RUNX1T1 subtype in the revised Introduction section. Our preliminary research found that KDM6A mutations predicted poor outcomes in patients with RUNX1::RUNX1T1, so we conducted a subgroup analysis on the RUNX1::RUNX1T1 subtype. This correlation suggests a potential biological interaction and clinical relevance, thereby justifying our focused investigation of this subgroup.
Revision made: We added this explanation to the last but one paragraph (line 62-63, with highlingt) of the Introduction to better justify the study design and highlight the significance of the RUNX1::RUNX1T1 subtype in the context of KDM6A mutations.
Comments 2: Discussion: Line 283-285: Please remove. The guidelines of how and what to write for discussion don’t need to be shown in Discussion.
Response 2: We agree with the reviewer. The noted sentence has been removed from the Discussion section. Lines 283–285 have been deleted in the revised manuscript.
Reviewer 2 Report
Comments and Suggestions for Authors
- What is the prevelance of AML among your cohort.
- How much patients acheive complete remission after chemotherapy, and what is the current relapse rate among your cohort.
- What is the mechanism of KDM6A in AML progression, how is it works.
- How you have calculate the sample size for this study.
- Is KDM6A also associated with other biomarkers such as FLT3, CEBPA.
- Measurable residual disease was identified using real-time quantitative polymerase chain reaction, please elaborate how you did this.
- Bioinformatics analysis methodology missing from methods section.
Author Response
|
Summary |
|
|
||||||||||||||
|
We sincerely thank you and the reviewers for your time and effort in evaluating our manuscript titled, “KDM6A variants increased relapse risk in adult acute myeloid leukemia” We are grateful for the opportunity to revise and resubmit our work to Cancers. The reviewers' insightful comments have been highly valuable in improving the quality and clarity of our manuscript. In the revised manuscript, we have addressed all the comments point by point. Below, we provide a detailed response to each issue raised by the reviewers. All changes in the manuscript have been highlighted using track changes for ease of review. |
||||||||||||||||
|
Point-by-point response to Comments and Suggestions for Authors |
||||||||||||||||
|
Comments 1: What is the prevelance of AML among your cohort. |
||||||||||||||||
|
Response 1: Thank you for pointing this out. We agree with this comment. Therefore, we have clarified in the Methods section that our study cohort includes 1,676 adult AML patients diagnosed and treated at our institution from January 2017 to July 2024. We highlight this explanation in line 71-72.
|
||||||||||||||||
|
Comments 2: How much patients achieve complete remission after chemotherapy, and what is the current relapse rate among your cohort. |
||||||||||||||||
|
Response 2: Agree. We have accordingly added information on complete remission (CR) and relapse rates in the Results section. Among 1,970 AML patients, 1,676 (85.1%) achieved final CR/CRi after induction chemotherapy. We added this explanation in lines 76-77 with highlight. With respect to relapse rate, KDM6A mutation had significantly higher 2-year cumulative incidence of relapse (CIR) compared with subjects with wild-type (50.9% [54.6%, 47.3%] versus 33.8% [33.7%,33.8%], P = 0.06 (Figure 3G).
Comments 3: What is the mechanism of KDM6A in AML progression, how is it works. Response 3: We have expanded the Discussion to briefly describe the functional role of KDM6A. KDM6A promotes AML progression and drug resistance through its tumor suppressor functions and involvement in DNA repair mechanisms. KDM6A is frequently mutated in AML, leading to loss of function and increased drug resistance[4]. For example, KDM6A mutations are associated with higher IC50 values for cytarabine, indicating reduced sensitivity to this common AML treatment. Additionally, KDM6A regulates the expression of DNA repair genes. Its deficiency impairs the DNA damage response (DDR), leading to increased sensitivity to PARP and BCL2 inhibition[3]. In AML cells, KDM6A is recruited to the transcriptional start sites of key homologous recombination genes upon DNA damage, facilitating their transcription. Loss of KDM6A activity, either through genetic or pharmacological inhibition, results in elevated H3K27me3 levels at these regulatory elements, preventing gene transcription and compromising DNA repair. This highlights KDM6A's essential role in maintaining genomic stability and its potential as a therapeutic target in AML. We highlight this explanation in line 333-345.
Comments 4: How you have calculate the sample size for this study. Response 4: the sample size was determined by sample size calculation for this retrospective study on KDM6A mutation and AML relapse, using 1:10 propensity score matching (PSM) was calculate as following: Key Parameters
Calculate required matched mutation-positive cases (k), using the cox model sample size formula for survival analysis:
Adjust for matching mutation prevalence.
To detect HR=2.0 with 80% power at α=0.05 using 1:10 PSM, collect minimum 1,438 subjects.
Comments 5: Is KDM6A also associated with other biomarkers such as FLT3, CEBPA. Response 5: We added a subsection in the Results describing the co-mutation analysis (line 210-211, with highlight). KDM6A mutations did not show significant co-occurrence with FLT3-ITD, FLT3-TKD, or CEBPA mutations in our cohort. (Figure 2C)
Comments 6: Measurable residual disease was identified using real-time quantitative polymerase chain reaction, please elaborate how you did this. Response 6: We expanded the Methods section to describe the MRD detection procedure. RT-qPCR was performed on BM samples using DNA extracted with DNAzol kits (Invitrogen) following standard protocols. Primers and TaqMan® probes for the target gene and internal control (ALB) were designed with Primer Express 2.0. Reactions were run on an ABI PRISM® 7500 using TaqMan® Universal PCR Master Mix, with specific pri-mer/probe concentrations and 150–250 ng DNA. Cycling conditions included an initial step at 50 °C for 2 min and 95 °C for 10 min, followed by 50 cycles of 95 °C for 15 s and 60 °C for 1 min. Gene expression was calculated by normalizing target gene copies to ALB. Detection sensitivity ranged from 10⁻⁴ to 10⁻⁵.[7] (line 92-100, with highlight).
Comments 7: Bioinformatics analysis methodology missing from methods section. Response 7: We added a detailed description of the bioinformatics analysis methodology in the Methods section (line 169-174, with highlight). |
||||||||||||||||

Reviewer 3 Report
Comments and Suggestions for Authors
Acute Myeloid Leukemia (AML) is often characterized by an aggressive course of the disease and by a heterogeneous genetic landscape, including gene point mutations, chromosomal translocations, deletion, and complex karyotypes. Altogether, they contribute to promoting leukemic cell proliferation, impairing cell differentiation and apoptosis, thus enhancing cell survival. As a consequence, AML is particularly challenging to treat. Mutations in the gene encoding KDM6A, a lysine demethylase family member, have been detected to different extents in several malignancies, including AML. Preclinical studies indicate that KDM6A variants are associated with worse survival in AML. However, the impact of KDM6A variants on the cumulative incidence of relapse in adults with AML in histological remission is untested. In AML, the translocation t(8;21) leading to the chimeric transcript RUNX1::RUNX1T1 is generally associated with a favorable prognosis.
The survey titled "KDM6A Variants Increased Relapse Risk in Adult Acute Myeloid Leukemia" aimed to investigate the correlation between KDM6A mutations and relapse risk in AML, with a specific focus on the RUNX1::RUNX1T1 subtype. The outcome of the study underscores that, though rare, KDM6A mutations in AML patients are recurrent and, when present in patients harboring the t(8;21), the predicted outcome is poor and the relapse risk is higher. Overall, the finding highlights the potential of epigenetic therapies for those patients displaying the t(8;21) alongside KDM6A mutations.
Overall, the manuscript does not suffer from major flaws, offers some interesting hints, and from my side, no concerns are raised.
Author Response
We thank Reviewer 3 for the positive evaluation and for highlighting the significance of our findings. No additional concerns were raised.